# Selective Inhibition of Pulmonary Vein Excitability by Constitutively Active GIRK Channels Blockade in Rats

**DOI:** 10.3390/ijms241713629

**Published:** 2023-09-04

**Authors:** Ian Findlay, Côme Pasqualin, Angèle Yu, Véronique Maupoil, Pierre Bredeloux

**Affiliations:** 1Laboratoire de Pharmacologie, Faculté de Pharmacie, Université de Tours, 37200 Tours, France; ian.findlay@univ-tours.fr; 2EA4245, Transplantation, Immunologie et Inflammation, Université de Tours, 37200 Tours, France; come.pasqualin@univ-tours.fr (C.P.); angele.yu@univ-tours.fr (A.Y.); veronique.maupoil@univ-tours.fr (V.M.)

**Keywords:** rat, cardiomyocytes, pulmonary veins, left atria, GIRK channels, tertiapin-Q, constitutive activity, electrophysiology

## Abstract

Pulmonary veins (PV) are the main source of ectopy, triggering atrial fibrillation. This study investigated the roles of G protein-coupled inwardly rectifying potassium (GIRK) channels in the PV and the left atrium (LA) of the rat. Simultaneous intracellular microelectrode recording from the LA and the PV of the rat found that in the presence or absence of acetylcholine, the GIRK channel blocker tertiapin-Q induced AP duration elongation in the LA and the loss of over-shooting AP in the PV, suggesting the presence of constitutively active GIRK channels in these tissues. Patch-clamp recordings from isolated myocytes showed that tertiapin-Q inhibited a basal inwardly rectified background current in PV cells with little effect in LA cells. Experiments with ROMK1 and KCa1.1 channel blockers ruled out the possibility of an off-target effect. Western blot showed that GIRK4 subunit expression was greater in PV cardiomyocytes, which may explain the differences observed between PV and LA in response to tertiapin-Q. In conclusion, GIRK channels blockade abolishes AP only in the PV, providing a molecular target to induce electrical disconnection of the PV from the LA.

## 1. Introduction

Ectopic activities arising from the cardiac sleeves of the human pulmonary vein (PV) are responsible for triggering atrial fibrillation [1]. Over the past few years, numerous studies have been conducted to characterize differences in the physiology of atrial and PV cardiomyocytes in an attempt to understand this phenomena (For review, [2]). In a rat PV-Left atria (LA) preparation, we showed that selective activation of α1-adrenergic receptors resulted in the abolition of electrical conduction within the PV, whereas electrical activity was maintained in the LA of the rat [3]. This effect was reversed by adenosine, which activates the G protein-coupled inwardly rectifying potassium (GIRK) channels [4,5], suggesting a role of these channels in PV electrical conduction.

Cardiac GIRK channels are composed of four subunits which are associated to form hetero-(Kir3.1/3.4) or homotetramers (Kir3.4/3.4) involved in the response of the atria to the vagal/parasympathetic nervous system activation [6]. Remodeling of atrial electrophysiology during the development of cardiac pathologies such as tachycardia and fibrillation have been found to include the development of constitutively active GIRK channels in man and a number of animal models [7,8,9,10]. The mechanisms leading to the conversion from GIRK activation by the βγ subunits of G-protein coupled receptors into spontaneous opening are still unclear [11,12,13,14].

In the atria, the peptide toxin tertiapin-Q (TQ), which selectively blocks the GIRK/Kir3.n channels family [15,16,17], reverses action potential (AP) shortening induced by acetylcholine (ACh). TQ induces also by itself atrial AP elongation in pathological [8,10,18] as well as in young and healthy animal models [19,20,21]. In consequence, GIRK channels could also provide a constitutively active background K^+^ conductance in the atria under physiological conditions.

This study was undertaken to compare the contribution of GIRK channels on AP in PV and LA cardiomyocytes. We found that TQ increased action potential duration (APD) in the LA, whereas a depolarization of the membrane inducing AP abolition was observed in the PV. Taken as a whole, these results suggest that GIRK channels may contribute more significantly to the regulation of constitutively active background K^+^ conductance in PV than in LA cardiomyocytes, offering a new pharmacological approach to inhibit the conduction of PV ectopic activity to the LA, triggering atrial fibrillation.

## 2. Results

### 2.1. Differential Effects of Tertiapin-Q upon the PV and the LA in Presence of Acetylcholine

The contribution of GIRK channels was investigated by studying the effect of TQ in presence of ACh on AP, simultaneously recorded at the frequency of 5 Hz in the PV and the LA of the same preparation (n = 5).

In the PV, 10^−7^ M ACh induced a slight but not statistically significant hyperpolarization (*p* = 0.071) with a significant (*p* = 0.025) increase in AP peak (Figure 1a,b,e). Unexpectedly, in the continued presence of Ach, the addition of 150 nM TQ led to a significant depolarization of PV from −76.6 ± 1.1 mV to −57.8 ± 1.2 mV (*p* < 0.001) and to the suppression of AP, which are replaced by small electrotonic waves (Figure 1c,e). In the LA, neither 10^−7^ M ACh alone nor ACh and 150 nM TQ had any significant effect upon DMP or AP peaks (Figure 1a–d). ACh alone reduced LA APD_90_ from 43.8 ± 1.4 ms to 35.9 ± 1.0 ms (*p* = 0.002). The addition of TQ then increased LA APD_90_ to 57.2 ± 3.0 ms (*p* < 0.001). This latter value was also significantly greater than the APD_90_, which had been recorded under basal conditions (*p* = 0.004).

These results indicate that 150 nM TQ had effects over and above the simple reversal of the effects of 10^−7^ M ACh. Traces in Figure 1c clearly show that, in the same preparation, TQ abolished AP in the PV, whereas they were still recorded in the LA.

### 2.2. Direct Effects of Tertiapin-Q upon the PV and the LA

We then examined the effects of TQ upon AP recorded simultaneously in the PV and LA of the same preparations in the absence of ACh (Figure 2, n = 10).

In these conditions, 150 nM TQ evoked a significant depolarization of PV from −74.1 ± 0.8 mV to −57.4 ± 1.7 mV (*p* < 0.001) associated with the replacement of AP by small electrotonic waves (Figure 2a–c). In the LA of the same preparations, TQ had no significant effect upon either the DMP (*p* = 0.173) or AP peak (*p* = 0.077) (Figure 2a,b,d), but it significantly increased LA APD_90_ from 45.7 ± 1.5 ms to 58.1 ± 1.3 ms (*p* < 0.001).

To investigate if TQ had direct effect upon the whole rat heart in our experimental conditions, its effect on sino-atrial node AP was studied. The application of 150 nM TQ alone had no effect upon spontaneous AP frequency (4.5 ± 0.01 Hz in basal condition versus 4.5 ± 0.1 Hz in TQ; *p* = 0.876), phase 4 duration (93.6 ± 6 ms in basal condition versus 90.4 ± 6 ms in TQ; *p* = 0.72), and APD_90_ (72.9 ± 2.9 ms versus 75.5 ± 3.6 ms in TQ; *p*= 0.636) recorded in 5 different preparations. On the other hand, TQ was able to significantly reverse the effects of 10^−6^ M of the muscarinic receptor agonist carbachol upon these parameters (Appendix A).

### 2.3. Reversal of Tertiapin-Q Effects upon the PV and the LA

#### 2.3.1. Reversal of Tertiapin-Q Effects by Acetylcholine

In the continuous presence of 150 nM TQ, we showed that 10^−7^ M ACh had no significant effect upon PV DMP or AP peak (Figure 3a,c; Table 1). Increasing the concentration of ACh to 10^−6^ M led to a significant hyperpolarization of PV (Table 1) and the restoration of over-shooting AP in 4 of the 7 preparations (Figure 3a,c). However, both PV DMP and AP peak were significantly less than those recorded in 10^−6^ M ACh under control conditions (Table 1). In the LA of the same preparations, while the different concentrations of ACh had no significant effect upon either DMP or AP peak (Figure 3b,d; Table 1), a concentration-dependent decrease of LA APD_90_ was observed (Figure 3b; Table 2), though LA APD_90_ was significantly greater than equivalent data obtained from control experiments in ACh alone (Table 2).

#### 2.3.2. Reversal of Tertiapin-Q Effects through Kir6.2 Channel Activation

We used the pinacidil analog P1075 to activate Kir6.2 (KATP) channels that are unaffected by TQ [17]. In PV, in the presence of 150 nM TQ, P1075 evoked a strong hyperpolarization from −56.8 ± 1.7 to −78.3 ± 1.0 mV (*p* < 0.001), and the recovery of over-shooting AP from the electrotonic waves at −54.0 ± 1.9 to 19.3 ± 0.4 mV (*p* < 0.001) in each of the 7 different preparations (Figure 4a,c). In the same preparations, P1075 had no effect upon either DMP (−74.7 ± 1.3 versus −76.3 ± 1.2 mV: *p* = 0.383) or AP peak (16.5 ± 1.2 versus 19.5 ± 0.8 mV: *p* = 0.063) in the LA (Figure 4b,d). P1075 significantly reduced LA APD under control conditions and in the presence of 150 nM TQ (Figure 4b; Table 3). In contrast to the results obtained with ACh (Table 2), there was only a slight difference between LA APD recorded in P1075 in the presence and the absence of TQ (Table 3).

Taken as a whole, these results show that TQ had a direct effect upon both the PV and LA of the rat, suggesting the existence of a constitutively active GIRK channels population in these tissues.

### 2.4. Selectivity of Tertiapin-Q against GIRK Channels in PV and LA Cardiomyocytes

The initial study describing the synthesis of TQ demonstrated that it had a high affinity for GIRK channels, whereas IRK1 (Kir2.1) channels were essentially insensitive to this molecule [22]. However, TQ was also shown to be effective against the Maxi-K^+^ (KCa1.1) channels [23] and ROMK1 (Kir1.1) [15], the latter being expressed in rat PV cardiomyocytes [24]. The Maxi-K^+^ channel blocker iberiotoxin [25] and the Kir1.1 channel blocker VU591 [26] were therefore tested upon the quiescent membrane potential of rat PV cardiomyocytes [27] and compared with a subsequent exposure to TQ (Figure 5a). Neither iberiotoxin (Figure 5b: *p* = 0.974) nor VU591 (Figure 5c: *p* = 0.698) had any significant effect. In both cases the subsequent application of TQ evoked a significant depolarization (*p* < 0.001 and *p* = 0.022, respectively). We conclude that, in the rat PV, the effects of TQ were not due to an action of the toxin on its other known molecular targets.

### 2.5. Constitutively Active GIRK Channels in PV and LA Cardiomyocytes

Finally the existence of a constitutively active ACh-activated K^+^ current (IK_Ach_) was studied in PV and LA using the whole-cell patch clamp technique on enzymatically isolated cardiomyocytes. The voltage clamp protocol consisted of a pre-pulse to −40 mV from the holding potential of −70 mV to inactivate the Na^+^ current. Then a voltage ramp from −120 to +60 mV at a rate of 18 mV/s which inactivated the Ca^2+^ current. I/V ramps were recorded in the absence of ACh, first under control conditions and then in the presence of 300 nM TQ. The currents blocked by TQ obtained by the subtraction I_control_–I_TQ_ in PV (red trace) and LA (black trace) in absence of ACh are illustrated Figure 6. In the PV cardiomyocytes, the TQ sensitive basal current showed an I/V curve with the characteristics of an IK_ACh_ [7]. It has a large inward component for membrane potentials below the potassium equilibrium potential (*p* = 0.029 versus LA at −100 mV). At potentials positive to E_K_, there is a clear outgoing current (*p* = 0.023 versus LA at −40 mV) which shows inward rectification. On the other hand, there was little or no TQ sensitive current in the cardiomyocytes of the LA

In an attempt to explain this difference between PV and LA cardiomyocytes, the expression of the GIRK1 (Kir3.1) and GIRK4 (Kir3.4) subunits of the cardiac GIRK channel were examined with the western blot technique (Figure 7). Both proteins responsible for the IK_ACh_ current are expressed in both PV and LA cardiomyocytes (Figure 7a). However, while the GIRK1 protein expression did not differ between the 2 tissues (*p* = 0.273 vs. LA, Figure 7b), the GIRK4 protein expression is 1.73 times greater in PV cardiomyocytes (*p* = 0.014 vs. LA, Figure 7c). This suggests the expression of GIRK channels composed of GIRK4 homo-tetramers in the PV as well as GIRK1/4 hetero-tetramers.

## 3. Discussion

The main finding of this study is that blocking constitutively active GIRK channels selectively abolishes excitability in the rat PV.

We found that TQ was able to reverse the effect of ACh upon LA APD and PV diastolic membrane potential beyond the values recorded in basal conditions, suggesting that it might have effects of its own in LA, but more particularly in PV cardiomyocytes. This was confirmed when we observed that TQ in the absence of ACh blocked an IK_ACh_ current and provoked a depolarization, leading to the reduction of AP to small electrotonic waves in PV cardiomyocytes, whereas it only induced an elongation of APD_90_ in LA, and had no effect on spontaneous AP of the sino-atrial node.

The fact that TQ was not having a generalized effect upon the heart is sustained by different studies. An APD elongation in the absence of ACh has been reported in dog [8,19] and guinea pig [20] LA and PV cardiomyocytes, whereas no effect was observed in ventricular APD [20] and the standard ECG parameters [16] in isolated guinea pig hearts. ACh and tertiapin were also devoid of effect in canine ventricular and purkinje cells [28]. In rat ventricles, TQ had no effect at the physiological stimulation frequency of 5 Hz but rather, prolonged APD and provoked a slight depolarization when the hearts were paced at 1 Hz [29]. Although functional, GIRK channels do not appear to have a significant impact on ventricular physiology or arrythmogenesis in mice [30]. In PV cardiomyocytes, TQ was reported to induce spontaneous AP in both the quiescent guinea pig PV [31] and the PV of rats with abdominal aorto-venocaval shunt [32]. These data, combined with the fact that we showed that TQ caused depolarization in quiescent rat PV without inducing spontaneous AP, support a selective loss of excitability in the cardiac muscle of the PV.

We interpret our results as arising from the inhibition of constitutively active GIRK channels by TQ, since they were obtained in the absence of Ach, partially reversed by the application of 10^−6^ M Ach, and fully reversed by the activation of Kir6.2 ATP-sensitive channels that are unaffected by TQ [17]. This suggests that they results from the selective blockade of GIRK channels and not from a non-selective blockade of multiple K^+^ channels, especially as TQ also has no significant affinity for Kir2.1 channels [17,22,33,34]. This was confirmed in control experiments with iberiotoxin and VU591, which ruled out the possibility that these results were due to the action of TQ on its other known targets i.e., Kir1.1 and KCa1.1. Finally, TQ in the absence of ACh reduced an IK_ACh_-like current in PV cardiomyocytes.

The differences observed in response to TQ between LA and PV cardiac muscle could be explained at least in part by the fact that the expression of GIRK4 is greater in PV than in LA cardiomyocytes, suggesting that GIRK4 homo-tetramers formation is more likely in PV than in LA cardiomyocytes. The absence of a differential reaction to TQ in the dog might then result from the equivalent distribution of GIRK1 and GIRK4 in the LA and PV [19,35]. While GIRK1/4 hetero-tetramers are also unaffected by intracellular Na^+^, the spontaneous activity of GIRK4 homo-tetramer channels increases with the intracellular Na^+^ concentration [12,36]. However, the link between the enhanced basal Na^+^ permeability of rat PV myocytes [37] and the effects of TQ upon PV membrane potential requires further investigation.

Another mechanism with similar electrophysiological effects as TQ upon the rat PV has been found to be α1-adrenergic receptor activation. Doisne et al. [27] first demonstrated a clear physiological distinction between the cardiac muscle of the LA and the PV of the rat. They showed that a combination of α1- and β1-adrenergic receptor stimulation evoked automatic electrical activity in the PV while having only a minor effect upon the membrane potential in the LA. They demonstrated that the biphasic response of the PV membrane potential to norepinephrine resulted from hyperpolarization associated with β1-adrenergic receptor stimulation and depolarization associated with α1-adrenergic receptor activation. Then, Bredeloux et al. [3] demonstrated that the depolarization induced by α-adrenergic receptor stimulation reduced PV AP peak to electrotonic waves, leading to the loss of electrical conduction from the LA along the PV. The GIRK channel agonist adenosine [5] induced PV membrane hyperpolarization and reversed these phenomena. A number of studies have demonstrated inhibition of GIRK channels with the α-adrenergic receptor agonist phenylephrine by means of PIP2 depletion in atrial cardiomyocytes [38,39,40,41]. In the same way, Deng et al. [42] showed that the chloride current blocker DCPIB reduced the activity of a number of the members of the Kir channel family, including GIRK1/4 hetero- and GIRK4 homo-tetramers, by interfering with PIP2 binding sites. Thus inhibition of constitutively active GIRK channels might explain the depolarization induced by α1-adrenergic receptors in rat PV cardiomyocytes [3,27] and the depolarization of the PV induced by DCPIB (Appendix A). TQ would evoke the same effects by direct block of the GIRK channel pore [34,43].

Application of longitudinal stretch to the rat PV also causes depolarization and a significant reduction of AP amplitude [44]. This phenomenon could be reversed by the chloride current blockers 4AC, DIDS, and DCPIB and has been linked to the activation of an I_swell_ chloride current in isolated myocytes from the rat PV. Since stretch was not applied to the atrium in those experiments, we do not know if this was specific to the PV, as it is the case for α-adrenergic stimulation [3,27] and here with TQ. We questioned whether PV depolarization induced by longitudinal stretch [44] and α-adrenergic stimulation [3,27] shared a common mechanism. Although ACh could reverse depolarization induced by stretch [45], neither DIDS nor DCPIB reversed depolarization induced by the α-adrenergic agonist cirazoline (Appendix A). Thus, we cannot explain the depolarization of the rat PV induced by α-adrenergic stimulation as resulting from the activation of an anionic current, while there are arguments in favor of GIRK channel inhibition.

In conclusion, all these data suggest that it is possible to induce electrical disconnection of the PV from the LA by one or more approaches such as α-adrenergic receptors stimulation [3,45], GIRK channel block with TQ, and activation of the I_swell_ anion conductance [44]. The common factor is the PV depolarization that leads to the reduction of AP peak and concurrent reduction of AP dV/dt_max_. This suggests a graded voltage-dependent inactivation of INa^+^ rather than direct block of conduction between LA and PV. The recovery of over-shooting AP in the PV by activation of GIRK (Figure 3) [3,45] and IK_ATP_ or block of I_swell_ [44] is associated with PV hyperpolarization that leads to the increase in AP amplitude and dV/dt_max_, suggesting recovery from inactivation of INa^+^.

Nevertheless, whatever the process, little effect is found upon the membrane potential of the LA wall. Since a significant difference in the quiescent membrane potentials of the PV and LA has been found [27], it may be proposed, therefore, that it is the regulation of the cardiac muscle membrane potential that underlies the responses of the PV and the lack of response of the LA wall. The regulation of membrane potential in the LA and the PV requires further investigation to identify molecular targets allowing the future development of more specific pharmacological treatments of AF. These could include constitutively active GIRK channels, whose blockade would prevent the conduction of ectopic electrical activity from the PV to the LA, as well as its remodelling and progression of the disease from paroxysmal to persistent form of AF. This could offer new therapeutic prospects for patients with paroxysmal AF who would not benefit from surgical pulmonary veins isolation (PVI) due to certain contraindications. Moreover, despite the high success rate of surgical PVI, arrhythmia recurrence related to PV reconnections are commonly observed [46,47] and require additional interventions. Thus, constitutively active GIRK blockade could also be an interesting adjunctive therapy by preventing functional electrical PV reconnection and arrhythmia recurrence, thus avoiding the need for new invasive procedures.

## 4. Materials and Methods

The protocols used in this study had been approved by the local Animal Care and Use Committee (Comité d’Ethique en Expérimentation Animale Val de Loire, Tours, France. Permit number 2022032119225830).

Male Wistar rats (CER Janvier, Le Genest-St Isle, France) were anaesthetized by intraperitoneal injection of pentobarbital (60 mg kg^−1^). After intravenous injection of heparin (500 IU kg^−1^), the heart and lung block was rapidly removed and placed in a dissecting dish that contained cold (+4 °C) cardioplegic solution (110 mM NaCl, 10 mM NaHCO_3_, 16 mM KCl, 16 mM MgCl_2_, 1.2 mM CaCl_2_, and 10 mM glucose).

### 4.1. Intracellular Microelectrode Recordings

The preparations used in this study consisted of either the LA and the superior PV, as previously described [3], or the right atrium and the superior vena cava. These were pinned out endocardial side up to the floor of an organ bath through which flowed Krebs-Henseleit solution (119 mM NaCl, 25 mM NaHCO_3_, 4.7 mM KCl, 1.18 mM KH_2_PO_4_, 1.17 mM MgSO_4_, 1.36 mM CaCl_2_, and 5.5 mM glucose; pH 7.4 equilibrated with 95% O_2_ and 5% CO_2_) at 5 mL/min and maintained between 35–36 °C.

Electrically evoked action potentials were recorded with either one or two glass capillary microelectrodes filled with 3 M KCl (20–30 MΩ) connected to a Duo 773 Electrometer amplifier (World Precision Instruments, Aston, UK). Voltage was filtered at 10 kHz low pass, displayed on an oscilloscope, and transferred at 40 kHz A/D via a PowerLab 4/25 interface (ADInstruments, Chalgrove, UK) to a PC computer running Chart 5.0 software. Electrical stimuli consisted of 2 ms duration square wave pulses generated by a Master-9 programmable stimulator (AMPI, Jerusalem, Israel) through a WPI A360 stimulus isolation unit and delivered to the left atrial wall of the preparation via two fine shielded Ag/AgCl wires. Depending on the preparations, evoked action potentials were recorded in either of the superior PV, as previously described [3].

Analysis of action potential parameters was performed with the Peak Parameters Module of Chart 5 software. The threshold for the action potential was set at >2 mV from the diastolic membrane potential.

### 4.2. Whole Cell Patch Clamp Experiments

Cardiomyocytes were enzymatically isolated as previously described [37,48].

Background potassium currents were recorded from PV and LA cardiomyocytes plated onto glass coverslips and maintained at room temperature (22–25 °C). During the patch clamp experiments, the cells were locally superfused by gravity with a Tyrode solution containing (in mM): NaCl 140; KCl 5.37; CaCl_2_ 1.36; MgCl_2_ 1; NaH_2_PO_4_ 0.33; HEPES 10; Glucose 10 (pH 7.4 with NaOH). The recording pipettes were pulled from borosilicate glass capillaries (Clark Electromedical Instruments, Edenbridge, UK) to a tip resistance of 2–4 MΩ. The pipette solution contained (in mM): DL-Aspartic acid K^+^ salt 115; KCl 25; MgCl_2_ 1; ATP-Mg 5; GTP Tris 0.1; NaH_2_PO_4_ 5; EGTA 10; HEPES 10 (pH 7.3 with KOH). An Axopatch 200 A amplifier connected to a PC computer, running Clampex (pClamp 9 software, Axon Instruments, Union City, CA, USA) through a Digidata 1200 A interface was used to control voltage and record currents from a holding potential (HP) of −70 mV. Data were acquired at a sampling frequency of 10 kHz and 2 kHz low-pass filtered with an 8-pole Bessel filter. Membrane capacitance (Cm) was measured by integration of the capacitive currents in response to a series of 10 hyperpolarizing pulses applied from the HP (amplitude and duration: 10 mV, 10 ms) and then averaged. The pipette and cell capacitances were compensated by 80%.

### 4.3. Preparation of Tissue Homogenates and Western Blotting

Tissue lysates were prepared from flash-frozen two pooled rat PV and LA. Samples were homogenized in lysis buffer (in mM: 30 histidine, 250 sucrose, 1× protease inhibitor cocktail cOmplete ULTRA (Roche Diagnostics, Singapore), 0.6 PMSF and 1 DTT) on ice. Insoluble material was removed by centrifugation at 11,000× *g* for 15 min, and the protein concentration of the supernatants was quantified according to the Bradford protein assay.

Proteins (30 µg) were then separated by SDS-PAGE on 9% tris-glycine gels and transferred to PVDF membrane (0.45 µm; Millipore, Burlington, MA, USA) using a wet transfer unit. After blocking of the membrane with 7.5% BSA in 0.1% TBS-Tween for 1 h at room temperature, the proteins of interest were labeled by the primary antibodies anti-GIRK1 and anti-GIRK4 (rabbit polyclonals, Alomone Labs, Jerusalem, Israel) overnight at 4 °C. The membranes were then incubated 1 h at room temperature with anti-rabbit HRP conjugated (Jackson ImmunoResearch, Ely, Cambridgeshire, UK) secondary antibody. The immunoblots were developed with Luminata Forte substrate (Millipore) and G: Box Chemi XR5 chemiluminescence imaging system (Syngene, Bengaluru, India). The protein-signal densities were normalized to the corresponding β-actin-signal densities. The analysis of the western blotting images was performed using ImageJ software 1.53f.

### 4.4. Statistical Analysis

Statistical analysis was performed by one way ANOVA, Student’s *t*-test, or Mann-Whitney test. Mood’s median test was used for western blot experiments. *p* values are given except for values less than 0.001 which are assigned a value < 0.001.

### 4.5. Chemicals

Chemicals were of reagent grade and obtained from either Merck KG (Darmstadt, Germany) or Sigma-Aldrich (Saint Quentin Fallavier, France). TQ and P1075 were obtained from Tocris Bioscience (Bristol, UK).

## Figures and Tables

**Figure 1 ijms-24-13629-f001:**
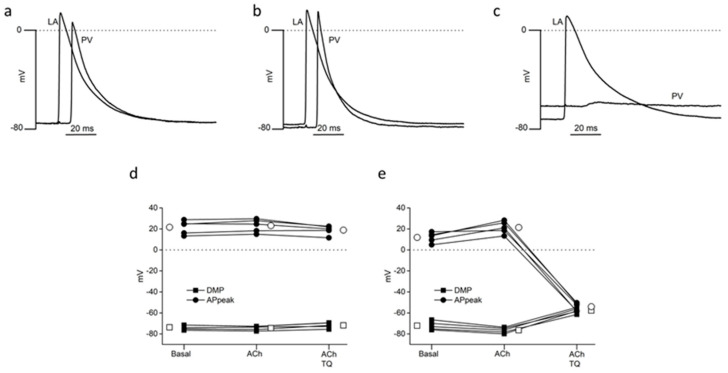
Effects of 10^−7^ M acetylcholine (ACh) and 150 nM tertiapin-Q (TQ) on electrically evoked (5 Hz) action potentials recorded simultaneously in the LA wall (LA) and the pulmonary vein (PV) of the same preparation. Pairs of individual traces (**a**–**c**) are from a continuous recording obtained under basal conditions (**a**), then during ACh superfusion (**b**), and finally in the presence of ACh with the addition of tertiapin-Q (**c**). (**d**,**e**) represent the results obtained from individual microelectrode penetrations in the LA and PV of five preparations, respectively. Squares represent diastolic membrane potential (DMP); circles represent action potential peak values (AP peak). Open symbols represent mean values. SEM are not visible on the figure because they are smaller than the symbols.

**Figure 2 ijms-24-13629-f002:**
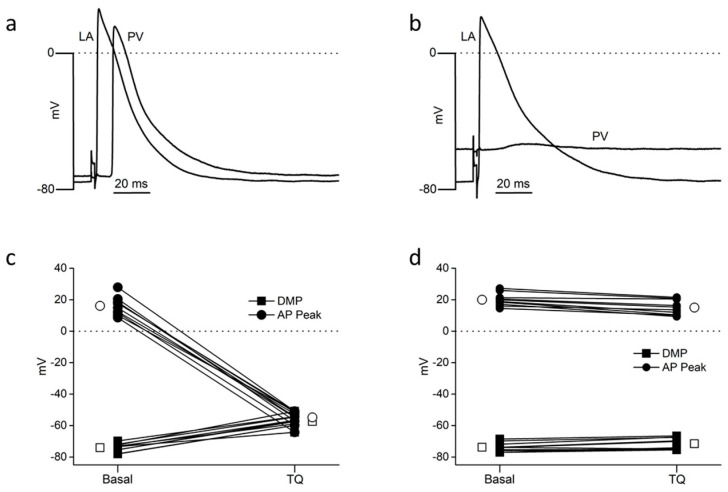
Effects of 150 nM tertiapin-Q (TQ) on electrically evoked (5 Hz) action potentials recorded simultaneously by intracellular microelectrodes placed in the LA wall (LA) and the pulmonary vein (PV) of the same preparation. Pairs of individual traces (**a**,**b**) are from a continuous recording obtained from one preparation under basal conditions (**a**), then during the superfusion of tertiapin-Q (**b**). (**c**,**d**) represent the results obtained from individual microelectrode penetrations in the PV and LA of ten preparations, respectively. Squares represent diastolic membrane potential (DMP); circles represent action potential peak values (AP peak). Open symbols represent mean values. SEM are not visible on the figure because they are smaller than the symbols.

**Figure 3 ijms-24-13629-f003:**
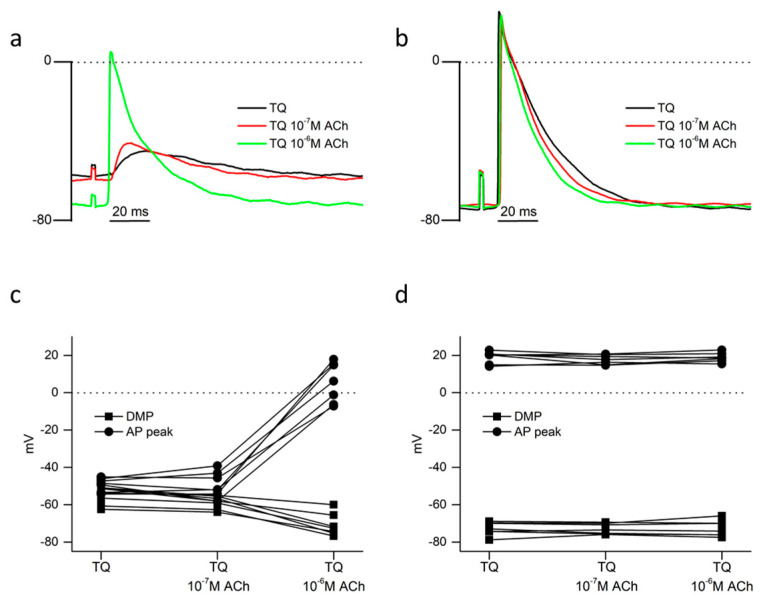
Concentration-dependent effects of acetylcholine (ACh) upon electrically evoked (5 Hz) action potentials in the PV and LA in the continuous presence of 150 nM tertiapin-Q (TQ). Traces are from a continuous recording obtained from one preparation in the PV (**a**) and LA (**b**). Data points linked by lines represent the results obtained from individual microelectrode penetrations in the PV (**c**) and LA (**d**) of 7 different preparations. Squares represent diastolic membrane potential (DMP); circles represent action potential peak values (AP peak). The mean and SEM values of the data are shown in Table 1.

**Figure 4 ijms-24-13629-f004:**
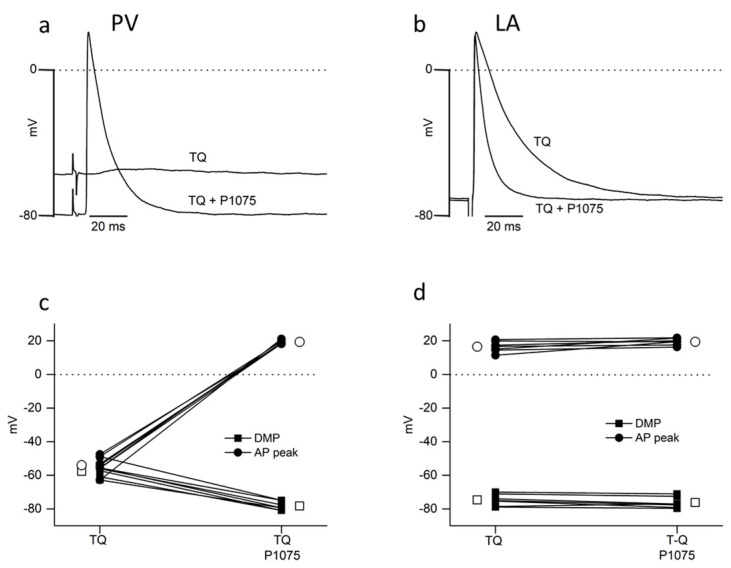
Recovery of over-shooting action potentials in the pulmonary vein (PV) by activation of Kir6.2 (KATP) channels with the pinacidil analog P1075 in the presence of tertiapin-Q (TQ). Electrically evoked (5 Hz) action potentials simultaneously recorded by intracellular microelectrodes placed in the LA wall (LA) and the PV. The pairs of individual traces are from a continuous recording in the PV (**a**) and the LA (**b**) in the presence of first 150 nM tertiapin-Q, then following the addition of 30 µM P1075. (**c**,**d**). Data points linked by lines represent the results obtained from individual microelectrode penetrations in the PV (**c**) and LA (**d**) of seven different preparations, respectively. Squares represent diastolic membrane potential (DMP); circles represent action potential peak values (AP peak). Open symbols represent mean values. SEM are not visible on the figure because they are smaller than the symbols.

**Figure 5 ijms-24-13629-f005:**
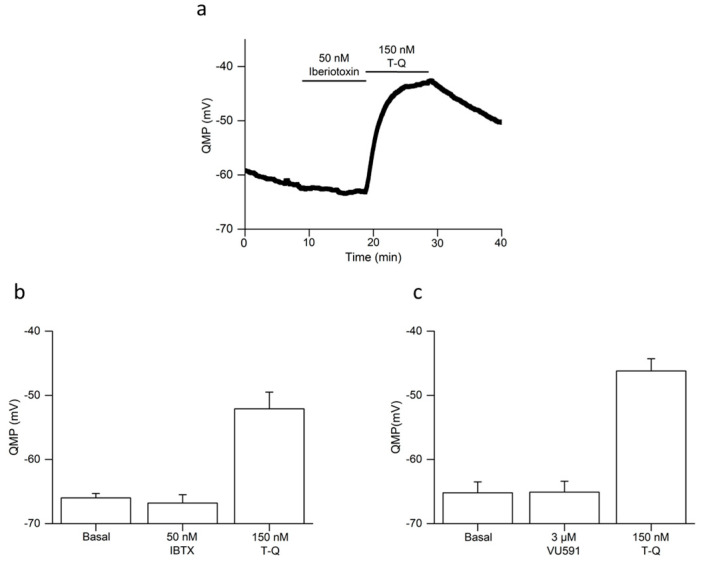
Comparison of the effects of iberiotoxin (IBTX), VU591, and tertiapin-Q (TQ) upon the quiescent membrane potential of cardiac muscle in the pulmonary vein (PV) of the rat. (**a**) Representative experiment comparing the effects of first iberiotoxin and then tertiapin-Q on the quiescent membrane potential (QMP). Data values for QMP were obtained by analyzing the otherwise continuous recording every second. The protocol consisted of that developed by Doisne et al. [27], where penetration of cardiac muscle in the PV was performed during 5 Hz electrical stimulation showing over-shooting action potentials. Electrical stimulation then ceased, and the preparation was allowed 10–15 min to establish an approximately steady-state QMP. In this example 10 min of superfusion of iberiotoxin was followed by 10 min superfusion of tertiapin-Q, as indicated by the bars above the recording. Columns and bars graphs represent mean and SEM values of QMP from 5 experiments comparing the effects of iberiotoxin and tertiapin-Q (**b**) and from 4 experiments comparing the effect of VU591 and tertiapin-Q (**c**).

**Figure 6 ijms-24-13629-f006:**
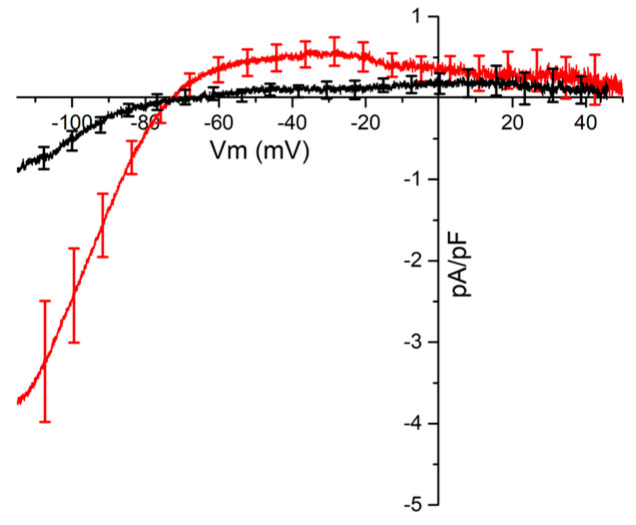
Tertiapin-Q (300 nM) sensitive currents in isolated left atria (black trace) and pulmonary vein cardiomyocytes (red trace) recorded under control condition in the absence of acetylcholine. Data are expressed as the means of voltage ramp obtained from 8 LA and 10 PV cardiomyocytes. For clarity, vertical bars representing SEM are not included for all data points.

**Figure 7 ijms-24-13629-f007:**
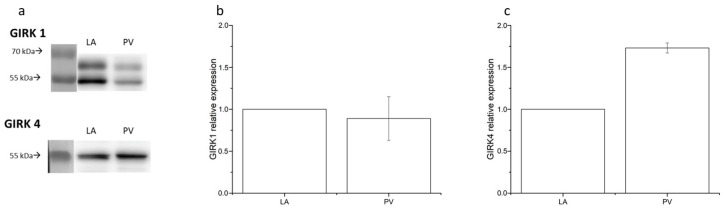
GIRK1 and GIRK4 subunits expression in the left atria (LA) and pulmonary vein (PV) cardiomyocytes. (**a**) Representative western blots of GIRK1 and GIRK4 in LA and PV cardiomyocytes. Bar graphs represent GIRK1 (**b**) and GIRK4 expression (**c**) in the PV relative to that in the LA. Data expressed as mean values ± SEM (n = 3, 2 rats/sample).

**Table 1 ijms-24-13629-t001:** Effects of acetylcholine (ACh) upon the PV and LA of the rat in the presence (n = 7) and the absence (n = 10) of 150 nM tertiapin-Q (TQ). Each preparation was stimulated electrically at 5 Hz. The data shown for tertiapin-Q represent the means ± SEM of the individual data points shown in Figure 3. *p* values in italics are given for statistical comparisons between data for diastolic membrane potential (DMP, in mV) and action potential peak values (AP peak, in mV) recorded under TQ and TQ + 10^−7^ M ACh, TQ + 10^−7^ M Ach, and TQ + 10^−6^ M ACh and TQ + 10^−6^ M ACh and 10^−6^ M ACh alone. The data for the effects of 10^−6^ M ACh alone were obtained from a separate series of experiments.

	TQ		TQ+ 10^−7^ M ACh		TQ+ 10^−6^ M ACh		10^−6^ M ACh
PV							
DMP	−55.3 ± 1.8	*p =* 0.145	−58.7 ± 1.3	*p* < 0.001	−70.9 ± 2.3	*p* = 0.012	−77.2 ± 1.0
AP peak	−49.2 ± 1.3	*p =* 0.988	−49.2 ± 2.5	*p* < 0.001	5.6 ± 4.0	*p* < 0.001	23.4 ± 1.3
LA							
DMP	−72.7 ± 1.3	*p =* 0.895	−72.9 ± 1.1	*p =* 0.959	−72.8 ± 1.6	*p* = 0.042	−76.3 ± 0.7
AP peak	19.0 ± 1.2	*p =* 0.417	17.7 ± 1.0	*p =* 0.401	18.9 ± 0.9	*p* = 0.471	17.7 ± 1.1

**Table 2 ijms-24-13629-t002:** Concentration-effect of acetylcholine (ACh) under control conditions or in the presence of 150 nM tertiapin-Q upon action potential duration (APD_90_, in ms) of the left atria electrically stimulated at 5 Hz. *p* values in italics are given for statistical comparisons of the basal, 10^−7^ M and 10^−6^ M ACh values are recorded under their respective experimental condition (control or tertiapin-Q). *p* values in bold are given for statistical comparisons between data obtained under control and tertiapin-Q condition in the absence or presence of ACh. APD_90_ values were obtained from experiments shown in Figure 3.

	Basal		10^−7^ M ACh		10^−6^ M ACh
Control (n = 10)	38.4 ± 0.9	*p <* 0.001	29.9 ± 1.2	*p <* 0.001	19.3 ± 1.3
	***p* < 0.001**		***p* < 0.001**		***p* < 0.001**
Tertiapin-Q (n = 9)	52.8 ± 1.6	*p* = 0.002	45.0 ± 1.0	*p =* 0.001	39.6 ± 1.0

**Table 3 ijms-24-13629-t003:** Effects 30 µM P1075 under control conditions or in the presence of 150 nM tertiapin-Q upon action potential duration (APD_90_, values in ms) of the left atria electrically stimulated at 5 Hz. *p* values in italics are given for statistical comparisons between data obtained with P1075 and the basal values recorded under their respective experimental condition (control or tertiapin-Q). *p* values in bold are given for statistical comparisons between data obtained under control and tertiapin-Q condition in the absence or presence P0175. APD_90_ values were obtained from experiments shown in Figure 4.

	Basal		P1075
Control (n = 9)	42.5 ± 2.1	*p <* 0.001	11.5 ± 1.2
	***p* < 0.001**		***p* = 0.020**
Tertiapin-Q (n = 7)	56.4 ± 2.1	*p <* 0.001	16.3 ± 1.5

## Data Availability

All data generated or analysed during this study are available from the corresponding author on reasonable request.

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
