# Peer review of "Selective Inhibition of Pulmonary Vein Excitability by Constitutively Active GIRK Channels Blockade in Rats"

_ijms, 2023, doi:10.3390/ijms241713629_

Round 1
Reviewer 1 Report
Dear Sir/Madam,
I had the opportunity to act as a reviewer on the recent submission by Findlay et al. to the International Journal of Molecular Sciences.
The authors present an interesting paper studying the role of the GIRK (G protein-coupled inwardly rectifying potassium) channels and their blockade as a potential molecular target to induce electrical disconnection of the pulmonary veins from the left atrium.
The manuscript is very well structured and written. However, some issues need to be addressed:
1. What are the effects of GIRK blockade on atrio-ventricular node and His-Purkinje system?
2. Are the effects of GIRK blockade across all pulmonary veins the same (according to the methods, only the junction LA-superior PV was studied).
3. On the translational aspect of the manuscript: the GIRK blockade could be a potential target for patients with atrial fibrillation (AF), primarily paroxysmal AF. However, for the paroxysmal AF the pulmonary vein isolation (PVI) is already proven to be a powerful tool. Therefore, the development of such treatment wouldn’t help much for paroxysmal AF – what are the author’s thought on GIRK blockade in persistent AF (i.e., adjunctive therapy after PVI): please expand in the Discussion.
4. GIRK is used as an abbreviation for G protein-coupled inwardly rectifying potassium – please use the same definition in the manuscript (and abstract – here the abbreviation is not even defined).
Best regards,
Author Response
Point-by-point response to reviewer 1 comments:
I had the opportunity to act as a reviewer on the recent submission by Findlay et al. to the International Journal of Molecular Sciences.
The authors present an interesting paper studying the role of the GIRK (G protein-coupled inwardly rectifying potassium) channels and their blockade as a potential molecular target to induce electrical disconnection of the pulmonary veins from the left atrium.
The manuscript is very well structured and written. However, some issues need to be addressed:
- What are the effects of GIRK blockade on atrio-ventricular node and His-Purkinje system?
This study was undertaken to compare the contribution of GIRK channels on action potential in PV and LA cardiomyocytes, i.e at the supraventricular level. That is why we did not investigate the effects of GIRK blockade on atrio-ventricular node and His Purkinje system although this could be an interesting subject.
In the literature, few articles have studied the effects of TQ on atrio-ventricular node. Drici et al., 2000 [16] found that in isolated unpaced guinea-pig heart, 300 nM tertiapin-Q per se did not change significantly the standard ECG parameters and in particular the PR interval. This suggest that blocking GIRK channels in the absence of acetylcholine does not affect the atrioventricular conduction.
Other papers have investigated the effect of TQ in vivo in genetically modified mouse models with sinus node dysfunction (Bidaud et al., Sci Rep. 2020 Jun 17;10(1):9835) or ex-vivo in myocytes isolated from the rabbit atrioventricular node using the patch-clamp technique (Choisy et al., Biochem Biophys Res Commun. 2012 Jul 6;423(3):496-502). Although TQ was able to improve atrioventricular conduction in vivo or to block the acetylcholine effects in isolated cells, the specific effect of TQ in the absence of an active parasympathetic stimulation was not described in these studies.
Finally, to our knowledge, only one study has investigated the effects of tertiapin on canine purkinje fibers. In this study, acetylcholine as well as tertiapin in the continuous presence of ACh were devoid of effect on Purkinje cells [28]. This differs from what we observed in PV cardiomyocytes, where TQ had effects over and above the simple reversal of the effects of acetylcholine.
This was now mentioned in the discussion line 295-296.
- Are the effects of GIRK blockade across all pulmonary veins the same (according to the methods, only the junction LA-superior PV was studied).
The tissue preparations used in this study are the same as those illustrated in the figure 1A published in Bredeloux et al., 2020 [3]. They consist of the right and left superior pulmonary veins and the atrium of the rat. Depending on the preparation, electrically evoked action potentials were recorded at multiple sites of the 2 superior PV and not only at the LA-superior PV junction. There were no differences in the effects of GIRK blockade between the two superior PV.
This is now specified in the methods on lines 400 and 413.
- On the translational aspect of the manuscript: the GIRK blockade could be a potential target for patients with atrial fibrillation (AF), primarily paroxysmal AF. However, for the paroxysmal AF the pulmonary vein isolation (PVI) is already proven to be a powerful tool. Therefore, the development of such treatment wouldn’t help much for paroxysmal AF – what are the author’s thought on GIRK blockade in persistent AF (i.e., adjunctive therapy after PVI): please expand in the Discussion.
We believe that targeting the myocardium of the pulmonary veins at an early stage by blocking the GIRK channels, could prevent the conduction of ectopic activity from the PV to the LA as well as its remodelling and progression of the disease from paroxysmal to persistent form of AF. This could offer new therapeutic prospects for patients with paroxysmal AF who would not benefit from surgical isolation of the pulmonary veins due to certain contraindications. Moreover despite the high success rate of surgical PVI, arrhythmia recurrence related to PV reconnections are commonly observed [47,48] and require additional interventions. Thus, GIRK blockade could also be an interesting adjunctive therapy, by preventing electrical PV reconnection and arrhythmia recurrence, thus avoiding the need for new invasive procedures.
This was now added at the end of the discussion.
- GIRK is used as an abbreviation for G protein-coupled inwardly rectifying potassium – please use the same definition in the manuscript (and abstract – here the abbreviation is not even defined).
Thank you for your comment. We have now used the correct abbreviation in the abstract, line 12, as well as in the introduction to the manuscript, line 35.
Best regards,
Reviewer 2 Report
I have enjoyed evaluating this manuscript. The authors make use of challenging, dual microelectrode impalements in an isolated rat atrial preparation to study the effects of superfused acetylcholine on atrial vs. pulmonary vein myocytes. This approach yields a novel and significant finding: ACh effects in the pulmonary vein are significantly different than those in the atrium and in fact, can result in an apparent lack of excitability in the pulmonary vein myocytes. This finding is the major strength of this paper and certainly constitutes the basis for publication of a revised paper. However, the authors should consider addressing some uncertainties in their original paper and also carefully rewriting and editing the Discussion to improve its clarity and hence the impact of this paper.
Major comments:
1. The pattern of results that is presented lead to the insight/conclusion that the resting potential at baseline of the atrial or PV myocyte is of critical importance. If so, the present of results and their discussion can and should be supplemented and clarified. My specific suggestions are:
- Adding a set of observations, made by applying current that would de- or hyperpolarize the preparation to directly test whether the exact value of the resting potential is an important variable. This could also be done by adding or subtracting potassium from the superfusing solution.
- More thoroughly reviewing and presenting previous findings on the actions and in particular on the selectivity of the actions of tertiapin-Q on baseline or background potassium conductances in the myocardium. I recall one or more previous publications reporting that this compound can block Kir2.1; and if so, this needs to be considered and discussed.
- To explain the primary mechanism of the interesting differential set of data that have been obtained, the authors invoke and provide some evidence for differential expression of GIRK channels in atrial vs. PV myocytes. This is plausible but another possibility is that the intrinsic input resistance of the PV myocytes is higher than that in atrial myocytes. This information should be able to be obtained from the data in Figure 6 but I was not able to convincingly extract it. In any case, this Figure is meant to convey important information concerning these background potassium conductances but the reversal potential seems to be approximately -75 mV which is not the Ek that would be expected from the potassium concentration that has been utilized.
- In general, the manuscript is carefully planned, well illustrated and quite clearly presented. However, this comment does not apply to the Discussion. This section needs to be rewritten and carefully edited so that it is less ambiguous and more clearly sets out key findings from the previous literature.
As noted above, this is a quite well planned and clearly presented manuscript. Nonetheless and importantly, the Discussion must be improved through rewriting, careful attention to clarification and removal of unconventional phrases; as well as additional and improved literature citations.
Author Response
Point-by-point response to reviewer 2:
- Adding a set of observations, made by applying current that would de- or hyperpolarize the preparation to directly test whether the exact value of the resting potential is an important variable. This could also be done by adding or subtracting potassium from the superfusing solution.
In fact, applying a current to the preparation via the intracellular microelectrode, although theoretically possible, does not work in practice. Indeed, the injected current rapidly decrease with distance in the LA or PV. This is due to i) the space constant, which depends solely on passive electrical characteristics of the layers of cardiomyocytes that form a syncitia in the LA or PV cardiac muscle and ii) the fact that the intracellular-extracellular resistance in cardiac muscle is asymmetric. We tried these experiments in a different context some years ago and in short, it was not possible to inject sufficient current to de-or hyperpolarize the tissue.
In a similar manner increasing the external concentration of potassium not only evokes depolarization but would also increase the conductance of both IK1 and GIRK, leading to a decrease in transmembrane resistance, a point to which we’ll return to later.
- More thoroughly reviewing and presenting previous findings on the actions and in particular on the selectivity of the actions of Tertiapin-Q on baseline or background potassium conductances in the myocardium. I recall one or more previous publications reporting that this compound can block Kir2.1; and if so, this needs to be considered and discussed.
To our Knowledge, all references assessing TQ selectivity against background potassium currents have shown that it has no significant affinity/effect upon Kir2.1 channels. For example, [22] were unable to determine a Kd value for tertiapin and tertiapin-Q against IRK1 (Kir2.1) channels. Similarly, Kitamura et al. 2000 [17] showed in rabbit cardiomyocytes, that tertiapin completely inhibited IKAch current with an IC50 of 8nM, whereas IRK1 (Kir2.1) channels were at least 100 times less sensitive to tertiapin. In this study, the authors showed that neither ATP-sensitive nor voltage-dependent potassium channels were affected by tertiapin. Ramu et al. 2004 [33], showed that TQ inhibited Kir2.1 channels expressed in xenopus oocytes at a Kd of 20 µM and in a structural simulation study Hilder and Chung (2013) [34] describe binding data with a Kd for TQ and Kir2.1 of 130 µM. This is a far cry from the TQ concentrations (150 and 300 nM) used in our study. It is therefore unlikely that our results are linked to the inhibition of Kir2.1 channels.
To our Knowledge, the only other channels that can be significantly blocked by TQ are those tested in our publication, i.e. the Maxi-K+ (KCa1.1) ion channels [23] and ROMK1 (Kir1.1)[15].
This is now specified in the text at line 218 in the result section and in the discussion line 309. The effect of TQ on cardiac muscle were also described in more detail in the discussion from line 292.
- To explain the primary mechanism of the interesting differential set of data that have been obtained, the authors invoke and provide some evidence for differential expression of GIRK channels in atrial vs. PV myocytes. This is plausible but another possibility is that the intrinsic input resistance of the PV myocytes is higher than that in atrial myocytes. This information should be able to be obtained from the data in Figure 6 but I was not able to convincingly extract it. In any case, this Figure is meant to convey important information concerning these background potassium conductances but the reversal potential seems to be approximately -75 mV which is not the Ek that would be expected from the potassium concentration that has been utilized.
Concerning Fig. 6. We are not sure that it would be possible to extract resistance data from these experiments since these are “difference” currents. Also, the data was obtained with a ramp voltage-clamp protocol. A square step protocol might be more appropriate, providing that there were no time-dependence. Concerning the reversal potential, largely we agree with you but these are very small currents and again the “difference” protocol meant that an accurate estimation – particularly in the LA was difficult to extract. It is curious but a number of references indicate reversal potentials for GIRK between -60 to -70 mV (Ehrlich et al 2003, 2004; Datino et al 2010; Koo et al 2010; Tsuneoka et al 2017). The main point however was that there was in essence no TQ-sensitive current in the LA cells while a clear, though not large inward and outward TQ-sensitive current was visible in the PV cells.
On the other hand, your comment about input resistance is a valid one. The relative resistance of the background current in LA and PV is one that we have considered for some years. A general point of comparison between the background K currents – usually IK1 – in atrial and ventricular myocytes indicates a much lower current in the atria, in particular for the outward part of the IV curve. The fact that a number of studies show the background K current is even less in PV than LA (Okamoto et al 2012; Tsuneoka et al 2017; Melnyk et al 2005; Arora et al 2007 [35]; Ehrlich et al 2003) would, as you suppose, predispose the tissue to large effects upon membrane voltage by quite small contributions of other currents. One could postulate that this might be the reason for the enhanced spontaneous GIRK – but this is really just a postulate and not something to put into a manuscript. The problem, particularly in a tissue rather than in isolated cells, is how to examine this experimentally. We do mention in the discussion that further studies are required to determine the mechanisms underlying the differences in the quiescent membrane potential in the rat LA and PV.
- In general, the manuscript is carefully planned, well illustrated and quite clearly presented. However, this comment does not apply to the Discussion. This section needs to be rewritten and carefully edited so that it is less ambiguous and more clearly sets out key findings from the previous literature.
We have extensively revised the Discussion.